# Opening Up a New Layer: A Deeper Look into "Interpreting CLIP with Hierarchical Sparse Autoencoders"

## Abstract

Sparse Autoencoders (SAEs) have become essential for decomposing model activations into interpretable concepts. However, despite their effectiveness, SAEs lack a natural ordering of features, making it difficult to prioritize important concepts under compute constraints. The Matryoshka Sparse Autoencoder (MSAE) was introduced as a means of learning nested subspaces, theoretically forcing high-level features into earlier dimensions to enable adaptive granularity. In this paper, we reproduce and analyze the MSAE framework. The reproduction of the study suffers from certain challenges, but the main claims still hold. Furthermore, this study extends the original paper by implementing Hyponym Activation Generalization as a metric, reducing computational cost and emissions with a method inspired by the Sandwich Rule, and deep-analyzing MSAE's ability to learn hierarchical information.

## 1 Introduction

Interpretability remains a fundamental challenge in machine learning, as the internal representations learned by models are often difficult to understand (Carvalho et al., 2019). Yet, interpretability is essential for uncovering the reasoning behind model predictions. Without it, validating, understanding, and ultimately trusting models becomes problematic. This concern is especially pronounced in high-stakes domains such as healthcare and law, where transparent and explainable decision-making is not only desirable but critical.

Sparse Autoencoders (SAEs) have proven to head in a promising direction to address the issue of polysemanticity for LLM interpretability (Shu et al., 2025). They have also emerged as a key technique for this by disentangling dense representations of vision-language models like CLIP into interpretable, sparse features (Zaigrajew et al., 2025). However, standard SAE approaches, such as ReLU or TopK SAEs, often struggle to balance reconstruction fidelity, how well the model preserves information with sparsity, and how interpretable the features are (Gao et al., 2024). This results in a rigid Pareto frontier in which improving one metric typically degrades the other (Rajamanoharan et al., 2024).

To address this limitation, Zaigrajew et al. (2025) introduce the Matryoshka Sparse Autoencoder (MSAE). Inspired by Matryoshka Representation Learning Kusupati et al. (2022), the MSAE architecture is designed to learn features at multiple granularities simultaneously. Around the same time, Bussmann et al. (2025) published a paper on MSAE independently, claiming that the hierarchical structure of this novel variant combats the issue of choosing a fixed number of latent activations for TopK SAEs. However, the difference between these two MSAE's variants is due to how they nest within the autoencoder. Zaigrajew et al. (2025) use a fixed dictionary that is trained across multiple dimensions using a sequence of increasing Top-k thresholds, while Bussmann et al. (2025) simultaneously train multiple nested dictionaries of increasing size.

By enforcing a nested structure on the latent space, where high-priority low-dimensional features are a subset of lower-priority high-dimensional ones, the model aims to prioritize semantically significant concepts while allowing for flexible inference at varying sparsity levels. The authors claim that this approach not only surpasses existing baselines in sparsity-reconstruction trade-off but also enables progressive recovery and more effective concept-based interventions. In this reproducibility study, we aim to verify these claims, focusing on the model's ability to recover hierarchical features and its utility in detecting and mitigating bias in downstream tasks.

Beyond verifying the original claims, this study introduces targeted extensions designed to evaluate the robustness and practical limitations of the MSAE framework. First, because hierarchical feature-splitting is highly vulnerable to feature absorption, we implement a Hyponym Activation Generalization metric to rigorously test the stability of MSAE's latent space against standard SAE baselines. Second, to address the immense compute overhead inherent to multi-granularity training, we introduce sandwich sampling to mitigate computational and emission costs without sacrificing structural performance. Finally, we leverage ImageNet's lexical hierarchy to evaluate whether MSAE's nested structure truly maps onto real-world concept granularities, bridging the gap between theoretical feature ordering and downstream interpretability.

## 2 Scope of Reproducibility

In this study, MSAE refers specifically to the formulation of Zaigrajew et al. (2025), therefore our conclusions apply only to Zaigrajew et al. (2025). We aim to verify the following claims:

**Claim 1:** MSAE achieves a superior trade-off between sparsity and reconstruction quality compared to existing baselines on CLIP embeddings.

**Claim 2:** MSAE actively organizes features from coarse to fine granularity, enabling gradual reconstruction.

**Claim 3:** MSAE discovers monosemantic, human-interpretable concepts in CLIP.

**Claim 4:** MSAE enables faithful concept-based interventions and bias analysis.

**Reproduction status.** Due to dataset and compute constraints, several experiments have been modified from the original experimental setting. We, therefore, distinguish between direct reproduction attempts, modified reproductions, and new extensions, which can be summarized in Table 4 in Appendix A.1. Claim 1 is evaluated using the same broad metrics as the original work but only at a single sparsity operating point per model, so it does not establish a full sparsity–fidelity frontier. Claim 2 is evaluated on the CC12M validation set along with the original ImageNet-1k, providing additional evaluation of gradual reconstruction. Claim 3 follows the original monosemanticity pipeline only partially, since we additionally report a relaxed threshold as a sensitivity analysis. Claim 4 reproduces the direction of concept-based interventions but does not include all causal controls needed to fully isolate SAE features.

## 3 Methodology

In this section, we discuss the models, datasets, and evaluation details. For replication, we use the public repository provided by the authors, which has code for pre-computing CLIP embeddings, training the SAE models, and evaluating post-training. Our adapted code can be found on GitHub (https://anonymous.4open.science/r/MSAE-sandwich-rep-1D0B/README.md).

### 3.1 Model Descriptions

### 3.1.1 CLIP

Contrastive Language Image Pre-training (CLIP) consists of a text and image encoder (Radford et al., 2021). The text encoder, which is a transformer-based model, encodes a sentence into a vector, while the image encoder, which uses a Vision Transformer, turns pixels into a vector. CLIP then uses contrastive learning to predict the correct pairings between images and text, pushing positive pairs together and negative pairs further away in the shared embedding space. It can operate in a zero-shot setting due to its ability to classify unseen images with an understanding of individual words and the visual patterns that relate to it, allowing for a robust model.

### 3.1.2 Sparse Autoencoders

SAEs are designed to learn interpretable feature representations from unlabeled data, rather than relying on manually engineered features (Ng et al., 2011). A standard autoencoder compresses data into a lower-

dimensional dense representation, whereas a SAE unembeds the data into a higher-dimensional vector but forces the network to activate only a small number of hidden neurons at a time, promoting sparse, interpretable feature representations. While standard autoencoders often use an undercomplete bottleneck, SAEs often use overcomplete hidden layers combined with a penalty term to enforce sparsity, resulting in better feature disentanglement and improved interpretability.

The SAE architecture leverages principles of dictionary learning. The columns of the decoder matrix can be viewed as features in a dictionary, where the hidden activations serve as sparse coefficients that decide which elements are necessary for each sample. SAEs reconstruct the input by applying a sparse weighted sum over these columns. Several architectures exist, such as ReLU, TopK, and BatchTopK SAEs. ReLU SAEs (Bricken et al., 2023) introduce sparsity in the reconstruction objective by applying $\ell_1$ regularization. However, this can cause activation shrinkage. TopK SAEs (Gao et al., 2024) combat this by relying solely on the $k$ highest activations for each sample, applying a strict sparsity constraint. BatchTopK SAEs (Bussmann et al., 2024) soften this constraint by applying the TopK operation on the entire batch. This ensures that each sample has a variable number of activations, but does not eliminate the need for a fixed top $k$.

### 3.1.3 Matryoshka Sparse Autoencoder

To allow SAEs to learn natural feature hierarchies, Zaigrajew et al. (2025) combined traditional TopK SAEs with Matryoshka representation learning (Kusupati et al., 2022) by simultaneously applying TopK operations with increasing values of $k$, $h$ times. This optimization across $k$ values allows for a nested structure, where high-level features captured by lower $k$ values form a subset of the more fine-grained features captured by higher $k$ values. The implementation by Zaigrajew et al. (2025) sets all values of $k$ as powers of 2 up to a dimension $d$, providing a balance between coverage and computational costs. For an input $x$, the latent feature vector $z$ and the reconstructed output $\hat{x}$ are computed at different granularity as:

$$z_i = \text{ReLU}(\text{TopK}_i(W_{enc}(x - b_{pre}) + b_{enc}))$$
$$\hat{x}_i = W_{dec}z_i + b_{pre}$$

for each $k$, with encoder matrix $W_{enc} \in \mathbb{R}^{n \times d}$, encoder bias $b_{enc} \in \mathbb{R}^d$, decoder matrix $W_{dec} \in \mathbb{R}^{d \times n}$, and preprocessing bias $b_{pre} \in \mathbb{R}^n$. The reconstruction objective is:

$$\mathcal{L}(x) := \sum_{i=1}^{h} \alpha_i ||x - \hat{x}_i||_2^2,$$

where $h$ represents the number of granularities and $\alpha_i$ are the weighting coefficients for a given granularity. The authors propose two configurations for the values of $\alpha_i$: Uniform Weighting (UW) and Reverse Weighting (RW). UW sets all $\alpha_i$ to 1, whereas RW gives more weight to lower $k$ values by setting $\alpha_i = h - i + 1$. During inference, any granularity of TopK operations can be applied and can also be omitted for the model to solely rely on ReLU activations. This flexibility removes the rigid number of active neurons from TopK SAEs, ensuring that the model can activate any number of features necessary to minimize the reconstruction error.

### 3.1.4 Concept Discovery

Rao et al. (2024) developed an automated neuron assignment method that leverages CLIP features encoded in a sparse high-dimensional latent space. The SAE decoder $\in \mathbb{R}^{M \times N}$ is defined as a learned dictionary $D = \{d_1, d_2, \ldots, d_M\}$, where each row vector $d_j \in \mathbb{R}^n$ represents a specific "neuron". We define a vocabulary $V = \{v_1, v_2, \ldots, v_K\}$ consisting of natural language concepts. For interpretability, each concept $v_k \in V$ is processed through the CLIP text encoder to generate the corresponding set of natural language embeddings, $E_{text} : t_k = \text{CLIPencoder}(v_k)$. For a specific neuron $j$, represented by dictionary vector $d_j$, we compute the cosine similarity between $d_j$ and the text embedding $v_k$:

$$Sim(j, k) = \frac{d_j \cdot t_k}{||d_j|| ||t_k||}$$

The neuron is assigned to a specific concept for the $j$-th neuron, defined by $v_{A(j)}$, which comes from the vocabulary, yielding the maximum alignment score:

$$A(j) = \underset{k \in \{1,...,K\}}{\arg\max} \; Sim(j,k)$$

## 3.2 Datasets

To reproduce the experiments and provide extensions of the paper, we use the following datasets: **CC12M** (Changpinyo et al., 2021), **ImageNet** (Russakovsky et al., 2015), **CelebA** (Liu et al., 2015), **Laion** (Schuhmann et al., 2021), **WordNet**(Fellbaum, 1998).

**CC12M** consists of roughly 12 million image-caption pairs from images and raw descriptions taken from the web. The original MSAE study trained on CC3M-scale data. In our study, we use CC12M to extract 3 million images to match the original training data amount while using a different data distribution. All SAE models are trained on CLIP ViT-L/14 embeddings extracted from this CC12M subset. This substitution is a deliberate modified-reproduction setting.

**ImageNet** is an image database organized according to the WordNet hierarchy, consisting of over 20,000 categories with on average 1000 images for each category. **ImageNet1k** and **ImageNet100** are subsets of ImageNet with 1000 and 100 categories, respectively.

**CelebA** is a large-scale dataset containing over 200,000 images of celebrity faces, annotated with 40 facial attributes. This dataset is used for the fourth claim where a single-layer classifier is trained on CLIP embeddings for binary gender classification. This revealed that concept magnitudes can increase the strength of concept-gender associations in a model.

**Laion** is a dataset with 400 million CLIP-filtered image-text pairs and their CLIP embeddings. These tokens were used to assign names to the neurons in experimentation.

**WordNet** is a lexical dataset that links labels from ImageNet to semantic relations and provides a hierarchical structure.

## 3.3 Metrics

**Sparsity rate** ($\hat{\mathbf{L}}_0$) is defined as the mean proportion of *zero*-valued elements in the SAE activations. It is not to be confused with the typical definition of $\mathbf{L}_0$, which is the number of *non-zero* activations.

**Fraction of variance unexplained (FVU)** measures reconstruction fidelity by quantifying how much of the input variance is not captured by the reconstruction (Gao et al., 2024). This is done by normalizing the mean squared reconstruction error $\mathcal{L}(x)$ by the mean squared value of the mean-centered input.

**Cosine Similarity (CS)** measures the similarity between two vectors by calculating the cosine of the angle between them.

**Linear probing (LP)** is used in the downstream task to evaluate how well semantic information is preserved in the reconstructed embeddings. A linear probe is trained on ImageNet100 using CLIP embeddings, and predictions from original and reconstructed embeddings are compared using two metrics: Kullback–Leibler (KL) divergence between predicted class distributions and classification accuracy.

**Centered Kernel Nearest Neighbor Alignment (CKNNA)** captures how well SAE activations align with the original input embeddings in kernel space by measuring representation alignment based on mutual nearest neighbors (Huh et al., 2024).

**Decoder Orthogonality (DO)** assesses the degree of orthogonality among monosemantic feature directions, by computing the mean cosine similarity over the lower triangular region of the SAE decoder weight matrix.

**Number of Dead Neurons (NDN)** indicates unused model capacity and a failure to learn meaningful semantic features by counting neurons that remain consistently inactive across all inputs during training.

**Hyponym Activation Generalization (HAG):** To investigate the hierarchical behavior of MSAEs versus standard SAE variants, we evaluate feature activation patterns across semantic hierarchies. When the dictionary size of an SAE scales, broader semantic features ideally split into fine-grained child features. However, as noted by Chanin et al. (2025), this process can suffer from "feature absorption," a failure mode in which a general latent is inappropriately suppressed and absorbed by its specific descendants. While true feature absorption requires per-instance probing (Karvonen et al., 2025), we introduce a new metric, which measures the ratio of mean activations of general, on selected features of hyponym text embeddings, which measures the degree of generalization across these splits. We denote this as Hyponym Activation Generalization (HAG). Further technical details, including baseline correlations and threshold selection, are provided in Section 4.5.1.

### 3.4 Models

All models are trained on CLIP ViT-L/14 embeddings of a 3M subset of CC12M CLIP embeddings with expansion factor 8 as the authors claimed it had the best performance. To keep it consistent, all models were trained for 30 epochs. The training duration and emissions can be found in Section 4.5.4.

- **ReLU**: Sparsity regularator $\lambda = 0.03$

- **BatchTopK**: Number of TopK neurons $k = 256$

- **MSAE (UW)**: Nesting list: [64, 128, 256, 512, 1024, 2048, 4096, 6144]

- **MSAE (RW)**: Nesting list: [64, 128, 256, 512, 1024, 2048, 4096, 6144]

- **MSAE Sandwich (RW)**: Nesting list: [64, 6144] + 1 random sample from [128, 256, 512, 1024, 2048, 4096]. See Section 4.5.3 for more details.

## 4 Experiments

We used the publicly available codebase by Zaigrajew et al. (2025) to reproduce the key claims in the study.

### 4.1 Claim 1: Sparsity-Fidelity Trade-off

To verify **Claim 1**, we assess the performance of various SAE models using the metrics explained above. All the SAE models were validated on the ImageNet-1k validation dataset, following the first table in the original paper. We report the mean and standard deviation of each metric across samples. CKNNA reports standard deviation over batches of 10,000 samples, while DO and NDN omit standard deviation, as these experiments were run on a single seed. HAG is measured for TopK=10 neurons and it's standard deviation is computed across WordNet categories (e.g., dog, fish, snake) from ImageNet100.

| Model | $\hat{L}_0\uparrow$ | FVU $\downarrow$ | CS $\uparrow$ | LP (KL) $\downarrow$ | LP (Acc) $\uparrow$ | CKNNA $\uparrow$ | DO $\downarrow$ | NDN $\downarrow$ | HAG (%) $\downarrow$ |
|---|---|---|---|---|---|---|---|---|---|
| ReLU ($\lambda = 0.03$) | $.811_{\pm.008}$ | $.293_{\pm.067}$ | $.951_{\pm.008}$ | $.148_{\pm.161}$ | $.904_{\pm.294}$ | $.626_{\pm.003}$ | 0.0025 | **0** | $31.39_{\pm16.01}$ |
| BatchTopK ($k = 256$) | $\mathbf{.890_{\pm.006}}$ | $.023_{\pm.027}$ | $.990_{\pm.009}$ | $.004_{\pm.006}$ | $.994_{\pm.075}$ | $\mathbf{.732_{\pm.003}}$ | 0.0024 | 9 | $\mathbf{8.79_{\pm7.57}}$ |
| MSAE (UW) | $\mathbf{.890_{\pm.009}}$ | $.077_{\pm.032}$ | $.971_{\pm.009}$ | $\mathbf{.001_{\pm.002}}$ | $\mathbf{.995_{\pm.073}}$ | $.727_{\pm.004}$ | **0.0019** | **0** | $11.51_{\pm9.81}$ |
| MSAE (RW) | $.797_{\pm.008}$ | $\mathbf{.004_{\pm.003}}$ | $\mathbf{.998_{\pm.001}}$ | $.003_{\pm.008}$ | $.988_{\pm.111}$ | $.705_{\pm.005}$ | 0.0022 | **0** | $13.16_{\pm9.61}$ |
| MSAE Sandwich (RW) | $.785_{\pm.009}$ | $\mathbf{.004_{\pm.002}}$ | $\mathbf{.998_{\pm.001}}$ | $.003_{\pm.006}$ | $.988_{\pm.110}$ | $.722_{\pm.003}$ | 0.0021 | 1 | $10.01_{\pm7.72}$ |

Table 1: Quantitative comparison of SAE models.

Table 1 shows similar trends to those seen in the original article with some notable deviation. MSAE (RW) performs better than MSAE (UW) in reconstruction metrics but not on $\hat{L}_0$, Linear Probing and CKNNA which deviates from the trends seen in the original paper. Similar to the original paper, we see that MSAE performs better than other models in reconstruction at the cost of sparsity. Interestingly, we see that the Sandwich technique for MSAE performs similar to MSAE (RW) model.

Another notable result that we see is BatchTopK's performance compared to MSAE in Hyponym Activation Generalization. While the original authors did not include any feature absorption related metric, Bussmann et al. (2025)'s variant included Chanin et al. (2025)'s method for feature absorption that indicated that MSAE outperformed BatchTopK in feature absorption. However, this variant of MSAE with flexible nesting structure over Bussmann et al. (2025)'s fixed nesting structure seems to suffer from increased hyponym absorption. Parts of this behaviour can also be seen in Section 4.5.2.

## 4.2   Claim 2: Gradual Reconstruction

For **Claim 2**, the authors used the metrics listed in Section 3.3 to perform progressive reconstruction using the TopK neurons while the $k$ value increased. The original authors validated this on ImageNet-1k. Additionally, we ran this experiment on the CC12M validation set of 190464 images to see if results similar to the original paper emerge.

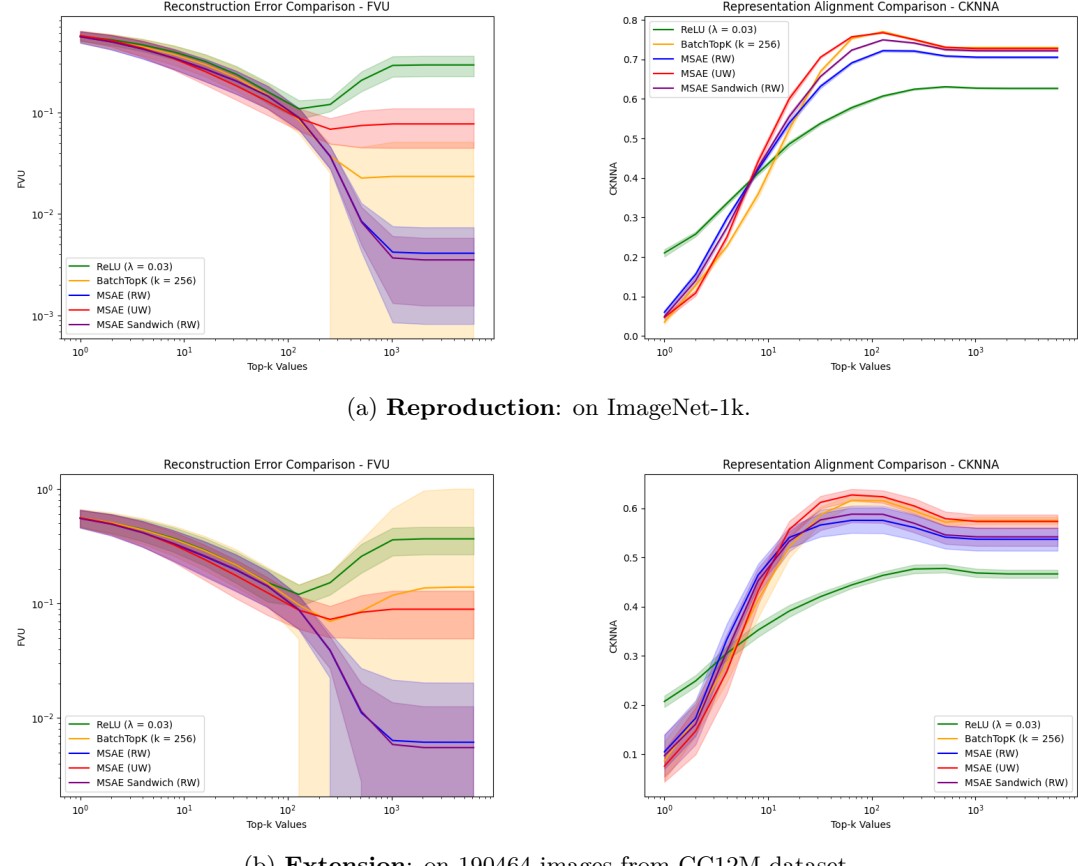

(a) **Reproduction**: on ImageNet-1k.

(b) **Extension**: on 190464 images from CC12M dataset.

Figure 1: Reconstruction (FVU) and Representation (CKNNA) metrics during progressive recovery runs for ReLU, BatchTopK, MSAE, and MSAE Sandwich variants across two datasets. Similar trends show up on both datasets, but CC12M subset has noticably higher variance.

In Figure 1, we see how different model reconstructions improve as we include more TopK neurons. For the FVU metric, in the reproduction case on ImageNet-1k, we see higher variance than the original paper and we also see new trends of ReLU's decreasing performance after using $10^2$ TopK neurons.

For representation alignment between the CLIP embeddings and SAE latent space, measured by the CKNNA metric, we see the ReLU model performs worse than in the original paper, where originally, it approached MSAE performance. This could likely be due to differences in hyperparameters from $\lambda = 0.003$ in the original

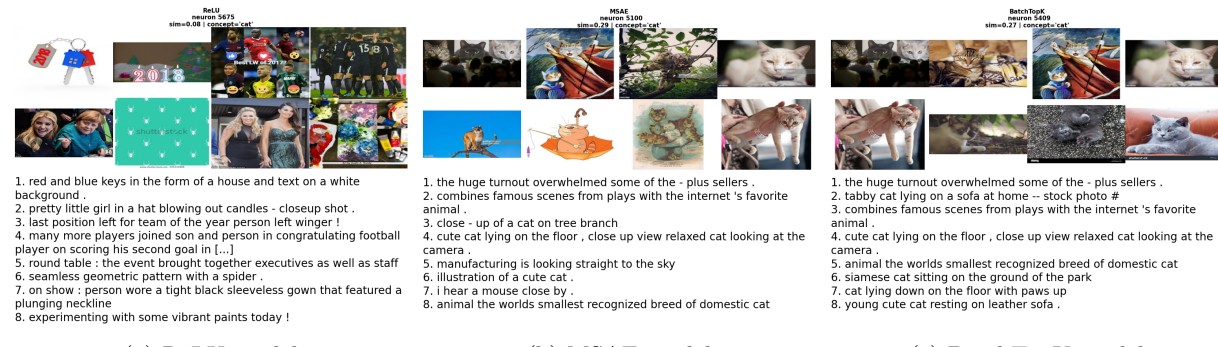

(a) ReLU model         (b) MSAE model         (c) BatchTopK model

Figure 2: Sparse version of the top-8 images and captions for the concept "cat" across different models.

paper, compared to $\lambda = 0.03$ in the reproduction. We also observed a loss in CKNNA performance after $10^2$ TopK neurons for BatchTopK and MSAE which was not present in the original paper.

Overall, we see that MSAE's general performance seems to match the original paper but with some deviations. We see a new trend of loss in CKNNA when all neurons are used. We also see worse performance in ReLU overall, which could be due to differences in hyperparameter, and a noticably higher variance across all models. The general trends hold up across new dataset as seen in CC12M, but with higher variance in this specific case.

## 4.3   Claim 3: Monosemanticity

To evaluate **Claim 3**, we examine whether individual SAE dictionary elements align with human-interpretable visual and textual concepts. The image-caption pairs are used for qualitative retrieval, while CLIP text embeddings from the LAION vocabulary serve only as an external set of concept labels.

For the quantitative analysis, we compute cosine similarities between the learned dictionary element and every concept embedding in the LAION vocabulary. A dictionary element satisfies the best-vector criterion when it is the highest-scoring element for at least one concept. We then count elements that both satisfy this criterion and exceed a specified similarity threshold.

We report our results under two similarity thresholds. The first, $\tau = 0.42$, reproduces the threshold used in the original study. Because this threshold yields few surviving neurons in our setting, we also report a percentile-matched threshold of $\tau = 0.26$. This threshold was obtained by matching the upper-tail percentile selected by the original threshold $\tau = 0.42$ in the reference similarity distribution to the corresponding percentile in our reproduced distribution. The calibration preserves the intended selectivity while accounting for differences in dataset scale, vocabulary, and dictionary dimensionality.

We additionally measure the ratio between the highest and second-highest concept similarities for each dictionary element. We treat this ratio criterion as diagnostic rather than as the primary selection rule because the highest-ranked concepts are often synonyms or closely related descriptions. A strict ratio threshold can therefore reject semantically coherent neurons whos top scores are distributed across several equivalent labels.

Table 2 shows that the original threshold of 0.42 produces a low yield selection across all models. The percentile-matched threshold $\tau = 0.26$ identifies substantially more candidate neurons while remaing selective. However, no neurons satisfy all conditions once the ratio criterion is enforced. We therefore use the similarity plus best-vector criterion as the main quantitative criterion and retain the ratio test as evidence that the original pipeline is sensitive to the exact concept vocabulary and similarity distribution.

For qualitative visualization, we retrieve the top-8 image-caption examples associated with selected neurons. We use two complementary retrieval procedures. In the dense probe, images and captions are ranked by their cosine similarity to the selected dictionary element. In the sparse probe, images are ranked by the

| Model | Similarity ≥ 0.26 \| ≥ 0.42 | Best vector | Above and best | Ratio ≥ 2.0 | All conditions |
|---|---|---|---|---|---|
| ReLU (λ=0.03) | 1169 \| 57 | 2789 | 1011 \| 57 | 14 | 0 \| 0 |
| BatchTopK (k=256) | 129 \| 9 | 1440 | 127 \| 9 | 3 | 0 \| 0 |
| MSAE (RW) | 271 \| 19 | 1711 | 265 \| 19 | 2 | 0 \| 0 |

Table 2: Number of neurons satisfying monosemanticity criteria across models under two similarity thresholds. The *best-vector* criterion counts unique dictionary elements that are the highest scoring element for at least one vocabulary concept. The *similarity + best vector* criterion requires a dictionary element to exceed the specified similarity threshold and to be the best-matching element for at least one concept.

activation of the corresponding SAE latent. The top-8 retrieval step is used only for visualization and is not treated as an additional neuron-selection criterion.

Figures 2 and 8 illustrate the distinction between these probes for the concept cat. Dense dictionary-based retrieval produces semantically coherent examples across all models. Sparse activation-based retrieval is more model dependent. In this case, ReLU retrieves less relevant examples, while MSAE (RW) and BatchTopK continue to retrieve examples that are visually consistent with the target concept. Across the additional examples reported in Appendix A.3, either MSAE or BatchTopK typically yields the more coherent sparse retrievals. These findings suggest that alignment in dictionary space does not necessarily imply equally selective sparse activation behavior.

Overall, our replication provides only partial support for Claim 3. Individual dictionary elements can align with coherent concepts, particularly for the MSAE (RW) model, but the quantitative yield depends strongly on the similarity threshold, the external concept vocabulary, and the ratio criterion. The results therefore support selective concept alignment rather than a robust conclusion that all evaluated SAE variants are uniformly monosemantic.

### 4.3.1 Bias Preprocessing

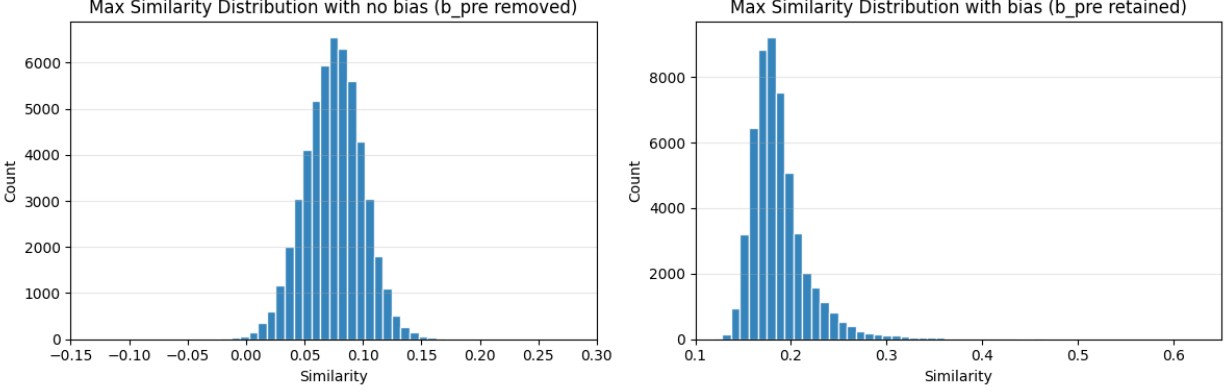

Figure 3: Effect of pre-activation bias preprocessing on neuron-concept similarity distributions.

To study the effect of pre-activation bias on monosemanticity, neuron-word similarities are evaluated using the LAION unigram and bigram datasets. The neuron activations are centered by subtracting the learned pre-activation bias prior to computing the similarities. This removes the offset that can lead to uniformly high similarity scores across many concepts, thereby obscuring neuron selectivity. The resulting similarity distributions show the impact of bias preprocessing on the emergence of monosemantic neurons.

The effects on neuron-concept similarity distributions is seen in Figure 3, based on the LAION 400m unigram dataset. When we omit bias preprocessing, cosine similarity scores cluster near 0.07, concealing the mean-

ingful neuron-concept relations. In contrast, incorporating bias preprocessing reveals a broader similarity distribution, restoring relations that are consistent with monosemantic behavior.

### 4.4 Claim 4: Concept-based Intervention and Bias Analysis

To verify **Claim 4**, we show that MSAE learns causally significant features used to detect and manipulate bias in downstream tasks. Specifically, we test whether intervening on specific features can directly alter the predictions of a gender classifier trained on CLIP embeddings. We first extract embeddings of CelebA images and attributes using CLIP ViT-L/14. We train a linear classifier on these embeddings, achieving an F1 score of 99%, the same as the paper.

We use a MSAE (RW) to decompose CLIP embeddings into sparse latent dimensions. We identify specific latent MSAE directions corresponding to semantic concepts often associated with gender bias ("bearded", "blonde", "glasses"). We perform the intervention by clamping the activation of the target feature $i$ to a fixed magnitude $\alpha$ ranging from 0 to 200. To maintain the model's sparsity constraint, if the target feature was not naturally active, we forced its inclusion by replacing the lowest-magnitude active feature. The edited latent $z_{edited}$ was decoded back into the CLIP embedding space $\hat{x}_{edited}$ and passed to the linear classifier. We record the shift in predicted probability for "Female" as a function of the intervention strength $\alpha$.

In Figure 4, increasing the activation of the features of "bearded", "blonde", and "glasses" systematically shifts the probability of the model to classify an image as "Male". Specifically, we observe an inverse relationship with the "blonde" neuron, increasing its activation reduces the "Male" probability, confirming that the linear probe associates blonde hair with "Female". In contrast, amplifying the "glasses" and "bearded" features increases the probability of a "Male" classification, a trend that is consistent with the paper. These results replicate the findings of the original study, confirming that MSAE successfully disentangles these concepts into distinct directions that align with the decision boundary of the linear probe.

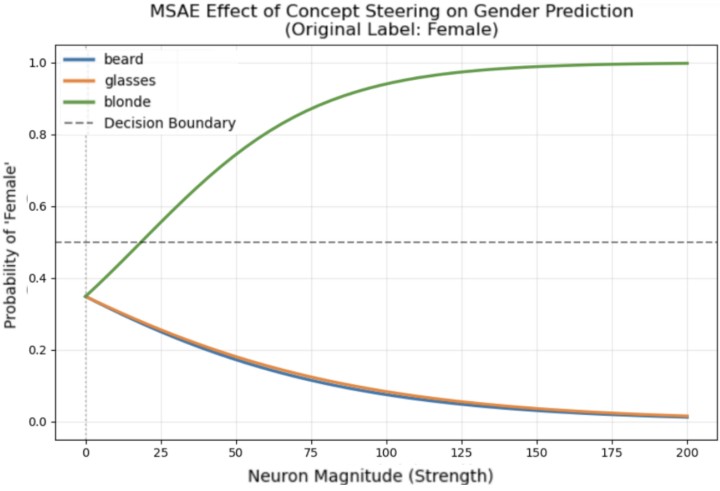

Figure 4: Increasing the concept magnitudes for "bearded", "glasses", and "blonde" in SAE space shows changes in the probabilities in gender classification in the CLIP space.

While the directional changes in classification probability support that SAE features align with meaningful semantic concepts, we further analyze intervention strength beyond the linear regime ($\alpha \to 200$). In this range, we observe a clear asymptotic saturation effect where the classifier's response plateaus. This suggests that after a certain activation magnitude, additional feature scaling does not produce proportional changes in the downstream prediction, indicating a bounded sensitivity of the linear probe in embedding space. Moreover, under the forced-inclusion protocol, we find that modifying a single sparse feature is often sufficient to induce measurable changes in prediction without destabilizing other latent dimensions. These results

highlight that SAE-based interventions can reliably steer downstream decisions, although their effect size is not unbounded and exhibits diminishing returns at high activation strengths.

### 4.5 Extensions

### 4.5.1 Hyponym Activation Generalization

To evaluate the robustness of SAE models on feature absorption, we compare activations induced by general concepts ("dog"), and specific concepts (dog breeds). These were collected by leveraging WordNet's hierarchical structure. The general concepts are a collection of synonyms and the specific concepts are hyponyms. Both sets of concepts are encoded using CLIP ViT-L/14 to obtain text embeddings, which are passed through the SAE encoder to produce latent activations. For general concepts, we compute the mean activation of each latent feature across concepts and select the top-k features with the highest average activation. These features are treated as general candidate features. We then measure the average activation of these same features under the specific concepts. The absorption score is defined as one minus the ratio between the mean activation of the selected features for specific concepts and their mean activation for general concepts. Intuitively, a higher score indicates that features strongly associated with general concepts fail to activate for more specific concepts, suggesting greater feature absorption.

### 4.5.2 Hierarchical Reconstruction on ImageNet

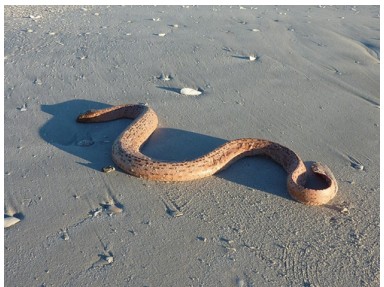

Figure 5: Image of a sea snake from node n01751748 with the hypernym path `entity -> physical_entity -> object -> whole -> living_thing -> organism -> animal -> chordate -> vertebrate -> reptile -> diapsid -> snake -> sea_snake`

To further investigate the hierarchical learning capabilities of Matryoshka latent representations, we make use of ImageNet's hierarchical labels. In Figure 5, we show a sample image from ImageNet along with the hierarchical labels extracted via its hypernym path.

For this experiment, we use a subset of ImageNet with 100 classes. The candidate label set includes all the labels in the hypernym path of these 100 classes. Since multiple hypernym paths are possible for the classes, we select only the longest hypernym path and break ties randomly. The classification labels at each level are found using cosine similarity between the reconstructions and candidate labels, similar to zero-shot classification using CLIP.

We measure the mean and standard deviation of *Accuracy* and *Level of Detail* using these classification labels across the samples in ImageNet100. For an image to be correctly classified, the label must be present in the hypernym path. *Level of Detail* is a measure of label abstractness between the range of $[0, 1]$, where 0 maps to the abstract labels (e.g. entity for Figure 5) and 1 maps to its most specific label (e.g. sea_snake for Figure 5). This is calculated using the label's index in the hypernym path divided by the length of the path. For correct guesses, the score is based on the image's hypernym path, and for incorrect guess, it is based on the mean *Level Of Detail* score of the classification label across all the hypernym paths. An ideal hierarchical model should always have high *Accuracy* and an increasing *Level of Detail* as it uses more neurons.

Figure 6 shows different model reconstructions in the classification task. The metrics were gathered for the following $k$ values: `[1, 2, 3, 4, 5, 6, 7, 8, 16, 24, 32, 48, 64, 82, 96, 128, 512, 1024, 2048, 4096, 6188]`. Overall, we see high variance across the models, but there are some interesting trends that appear based on the mean scores.

In terms of accuracy, we see that the ReLU model has an increasing accuracy as more neurons are used. However, BatchTopK and all MSAE models show high accuracy when using only 1 neuron, then the accuracy

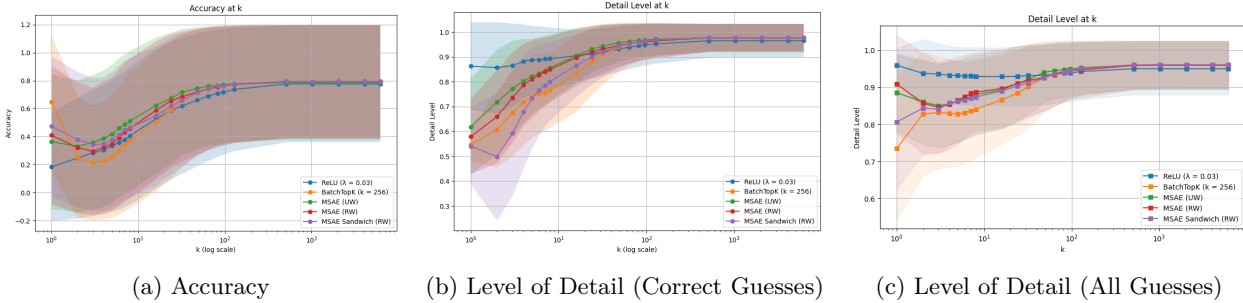

(a) Accuracy  (b) Level of Detail (Correct Guesses)  (c) Level of Detail (All Guesses)

Figure 6: Accuracy and Level of Detail at different levels of reconstruction ($k$) for ImageNet-100. All models have high variance. Based on mean scores, BatchTopK and MSAE Sandwich seems to build features from 'coarse' to 'fine' whereas MSAE (UW/RW) typically only does that for correct guesses. ReLU typically tries to directly reconstruct the final label.

drops down and starts increasing again. There is a difference in strength of this trend across the models with BatchTopK showing strong signs, while it is less pronounced in all MSAE variants.

These trends can be partly explained by the *Level of Detail* metric. We see that ReLU tends to always predict the final class label regardless of the number of neurons used. Hence, it becomes more accurate as more neurons are used. However, BatchTopK and MSAE models show a trend of building up the reconstruction from 'coarse' to 'fine' features in terms of hypernym path labels.

Interestingly, this is not the case for MSAE (UW/RW). There seems to be a behavior of the MSAE model trying to directly reconstruct a final label with it's first TopK neuron. This could be due to the Matroyshka's weights in the loss function incentivizing the model to directly predict the final labels with just a few neurons if possible. This is further supported by the fact that MSAE Sandwich shows traits of BatchTopK and it has reduced influence of the Matroyshka's weights in the loss function due to the Sandwich sampling method only using 3 granularities.

The drop in accuracy after using more than 1 neuron, while increasing the *Level of Detail* could be due to the increase in possible candidate label sets for that *Level of Detail*. For example, the first deviation on the Hypernym path for this dataset only has 3 possible candidates [chordate, invertebrate, game] but going 3 levels down, there are 32 possible candidates. In Appendix A.4, we provide *Accuracy*, *Level of Detail* and model predictions plots for randomly selected classes from ImageNet100.

Overall, we see that ReLU models do not reconstruct in a hierarchical manner. BatchTopK implicitly reconstructs hierarchically, but MSAE slightly deviates from hierarchical reconstruction due to its $k = 1$ behavior. MSAE with Sandwich sampling does not face the same issue, suggesting that strict, regular granularization of latent space might not be ideal for hierarchical learning.

### 4.5.3 Sandwich sampling

To reduce the computational cost of training a MSAE model, we apply a method inspired by the sandwich rule introduced by Yu and Huang (2019). The authors of this paper improved the efficiency of universally slimmable networks (US-Nets) by training the model at smallest width, largest width, and $(n - 2)$ random widths at each iteration. To make this method applicable for MSAEs, we compute the reconstruction loss for a subset of granularities. The smallest granularity guarantees robust core features, the largest granularity ensures full model capacity, and a randomly sampled middle layer retains the MSAE hierarchy. In addition to this, we also experiment with using only random granularities. More information on ablation can be found in A.5. The resulting loss is scaled up by the fraction of total layers to sampled layers to ensure that the magnitude of the gradient is consistent with the original MSAE implementation. We train the MSAE with sandwich sampling using the same hyperparameters used for the other MSAE models.

Table 1 reveals that MSAE (RW) with sandwich sampling has almost the same performance as MSAE (RW) while requiring approximately 24% less energy during training as shown in Table 3. It also has competitive

scores on all other metrics except sparsity. However, ReLU and BatchTopK still beat all MSAE variants in terms of emission and training time.

### 4.5.4 Emission tracking

We used codecarbon (Lottick et al., 2019) to track the emissions for training different SAE models. We focus on the following metrics: emissions, cpu energy, gpu energy, ram energy, and energy consumed. All models were trained on a single A100 GPU.

| Model | Duration (HH:MM:SS) | Emissions [$CO_2$eq] (kg) | Energy (kWh) | | | |
|---|---|---|---|---|---|---|
| | | | CPU | GPU | RAM | Total |
| ReLU ($\lambda = 0.03$) | **01:17:01** | **.064** | **.036** | **.192** | **.012** | **.241** |
| BatchTopK ($k = 256$) | 01:19:50 | .067 | .038 | .200 | .013 | .252 |
| MSAE (UW) | 01:50:17 | .110 | .053 | .341 | .018 | .411 |
| MSAE (RW) | 01:50:54 | .108 | .054 | .330 | .018 | .401 |
| MSAE Sandwich (RW) | 01:32:49 | .082 | .044 | .247 | .015 | .307 |

Table 3: Emissions during training SAE models.

Table 3 shows that MSAE models are the most computationally expensive and ReLU the least. MSAE with sandwich sampling outperforms the other MSAE models with approximately 24% less energy usage. Table 7 in Appendix A.6 shows the performance versus training emissions across all model variants.

## 5 Discussion & Conclusion

Through our experiments, we show that several of the main claims introduced by Zaigrajew et al. (2025) are reproducible with some minor deviations. The architectural and reconstruction-related finding are broadly supported, the MSAE model achieves strong reconstruction and representation-alignment performance at the evaluated operating point. However, the evidence for Claim 3 on monosemanticity is less conclusive. Its strict validation criteria are sensitive to the similarity threshold and concept vocabulary and do not generalize robustly to our reproduction setting. Thus, while our results support the core structural advantages of MSAE, they provide only partial support for its semantic interpretability claims. To address the limitations found in Claim 3, we introduce an alternative evaluation criterion that removes the strict ratio criterion. This modification accounts for cases in which several near-synonymous or closely related labels receive similar scores for the same feature. The resulting criterion offers a less brittle approach to identifying candidate monosemantice features.

We further extend the original work through experiments designed to improve our understanding of model interpretability and efficiency. First, we introduce the Hyponym Activation Generalization metric to gain more insight into model monosemanticity. Here, we see that BatchTopK is the best model, followed by the different MSAE variants. Moreover, we conduct further experiments to examine whether the features are reconstructed hierarchically. Specifically, we investigate whether low-granularity features represent abstract concepts and if increasing granularity gradually adds finer details. Our findings indicate that ReLU is not able to reconstruct features in a hierarchical manner, whereas BatchTopK can implicitly and MSAE counter-intuitively tends to reconstruct with fine details with the 1st TopK neuron, but then starts to reconstruct features in a hierarchical manner with the addition of more neurons. Lastly, MSAE with sandwich sampling presents competitive performance in many evaluation metrics while reducing emission costs compared to the original MSAE model. In the future, we hope to further analyze and evaluate the sandwich method by finetuning all the hyperparameters.

Overall, our results show that MSAE remains competitive compared to the other baseline models, particularly in terms of reconstruction as introduced by Zaigrajew et al. (2025). However, BatchTopK still remains a highly competitive model and performs best under our Hyponym Activation Generalization metric. Taken together, the findings support the structural and efficiency advantages of MSAE more strongly than the claim that its individual features are consistently monosemantic.

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

# A    Appendix

## A.1    Scope of Reproducibility

We provide a table summarizing direct reproduction attempts, modified reproductions, and new extensions.

| Experiment | Original setting | Our setting |
|---|---|---|
| Claim 1 | Sparsity–fidelity comparison | Single operating point per model |
| Claim 2 | ImageNet-1k gradual reconstruction | ImageNet-1k & CC12M validation gradual reconstruction |
| Claim 3 | Original thresholding pipeline | Threshold sensitivity analysis on CC12M setting |
| Claim 4 | Concept intervention | Forced-inclusion feature intervention |
| HAG | Not in original paper | Exploratory extension metric |
| Hierarchical reconstruction | Not in original paper | Exploratory ImageNet/WordNet evaluation |
| Sandwich sampling | Not in original paper | Exploratory efficiency extension |

Table 4: Summary of which experiments are direct reproductions, modified reproductions, and exploratory extensions.

## A.2    Gender Bias Analysis in CelebA

Consistent with the original framework, we analyze gender bias via a CLIP-based linear classifier trained on the CelebA dataset. We validate the hypothesis that distinct latent concepts exhibit strong gender-specific associations by performing evaluations across both female (Figure 4) and male classes (Figure 7).

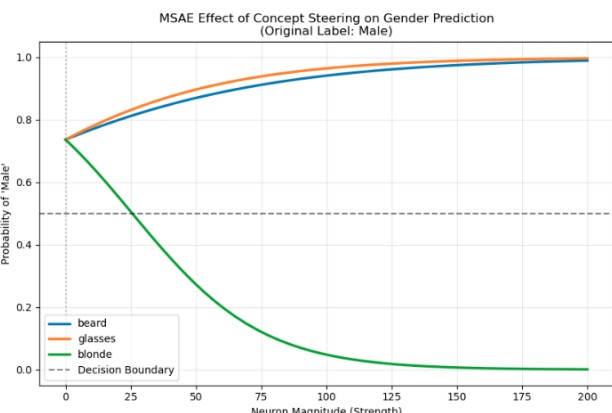

Figure 7: We showcase the male association of the words "bearded", "glasses", and "blondes". This follows the original paper as it behaves exactly opposite of the female case.

### A.3 Monosemanticity

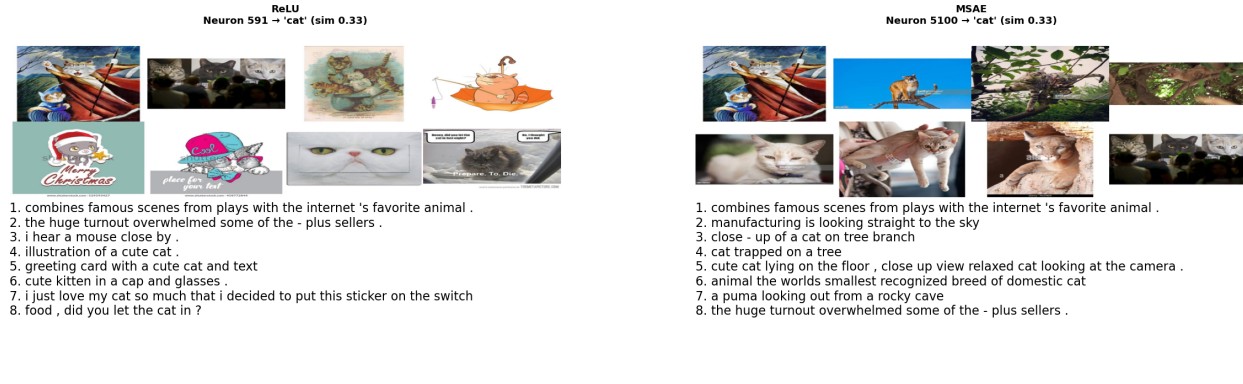

(a) ReLU model  (b) MSAE model

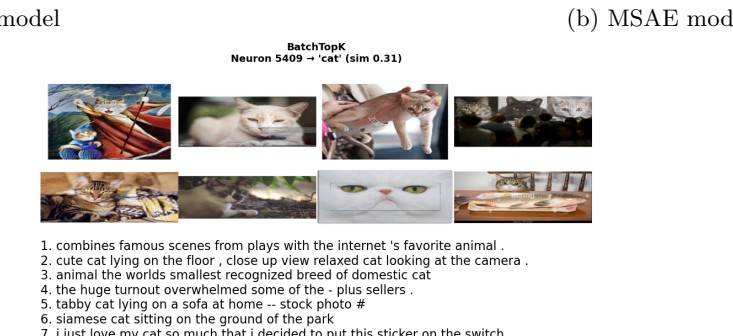

(c) BatchTopK model

Figure 8: Additional Top-8 images and captions for the concept "cat".

Figure 8 presents the top-8 images and captions associated with the concept "*cat*" across different models. The models show monosemanticity, confirming neurons successfully align with specific concepts

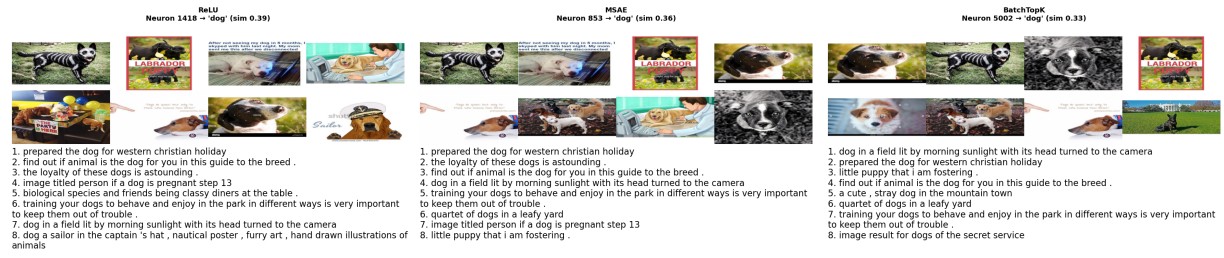

| (a) ReLU model | (b) MSAE model | (c) BatchTopK model |

Figure 9: Dense version of the top-8 images and captions for the concept "dog", where all models retrieve relevant images.

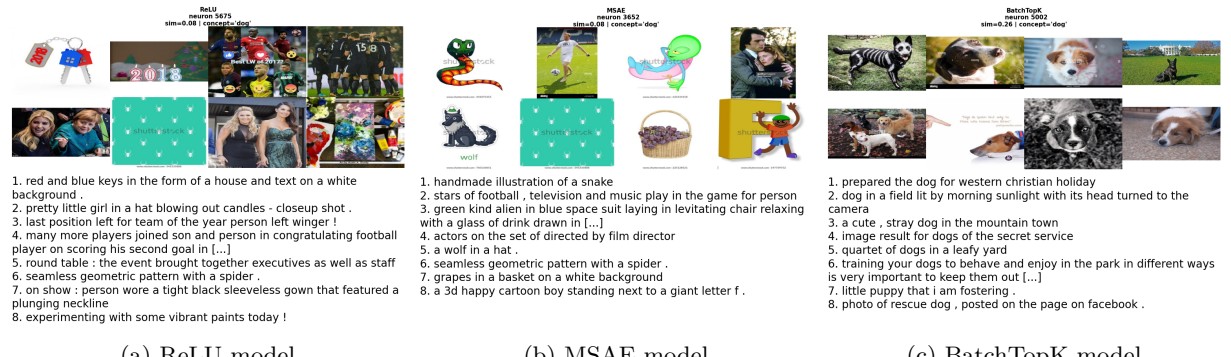

| (a) ReLU model | (b) MSAE model | (c) BatchTopK model |

Figure 10: Sparse version of the top-8 images and captions for the concept "dog", only BatchTopK retrieves relevant images.

Figures 9 and 10 show that all models retrieve relevant images in the dense setting, whereas retrieval quality in the sparse setting is more model-dependent. For the dog concept, BatchTopK retrieves the relevant examples, while for the bear concept in Figure 11, MSAE retrieves the more coherent examples again.

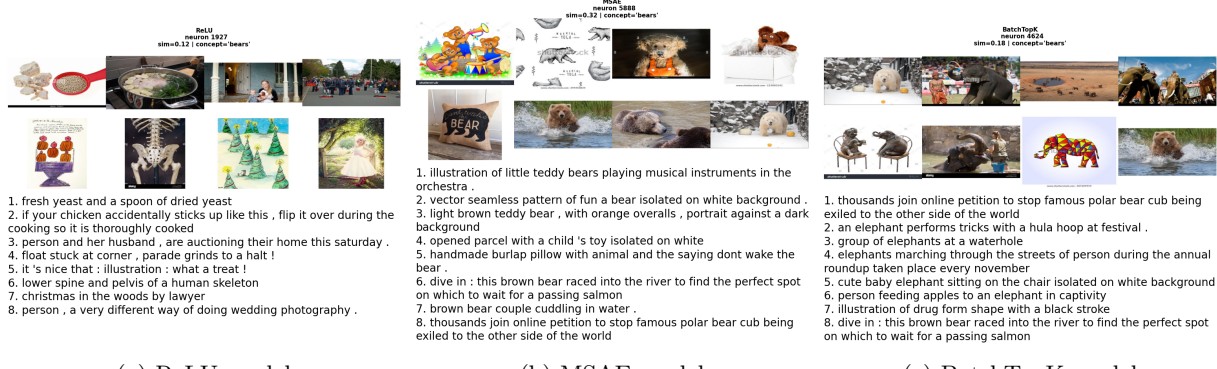

| (a) ReLU model | (b) MSAE model | (c) BatchTopK model |

Figure 11: Sparse version of the top-8 images and captions for the concept "bear", only MSAE retrieves relevant images.

### A.4 Accuracy and Level of Detail for 3 random classes in ImageNet100

Below are the Accuracy and Level of Detail plots for 3 randomly selected classes from ImageNet100. For certain classes, there were no correct classification for certain $k$ values, leading to gaps in the Level of Detail plots.

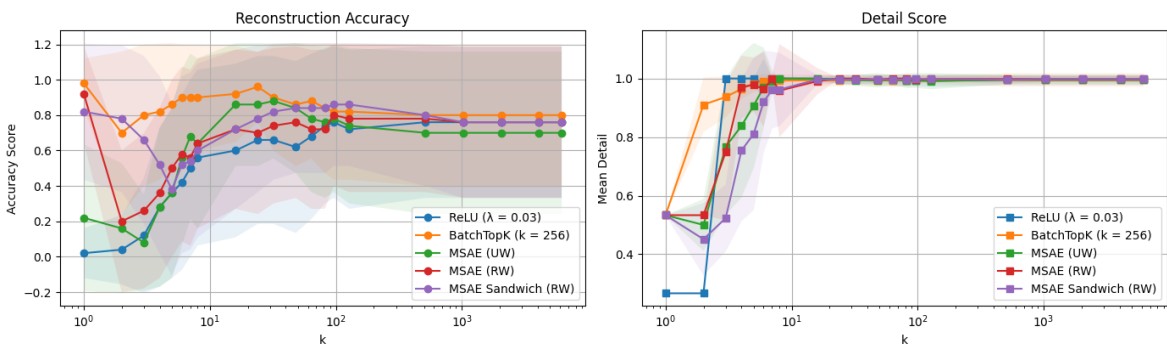

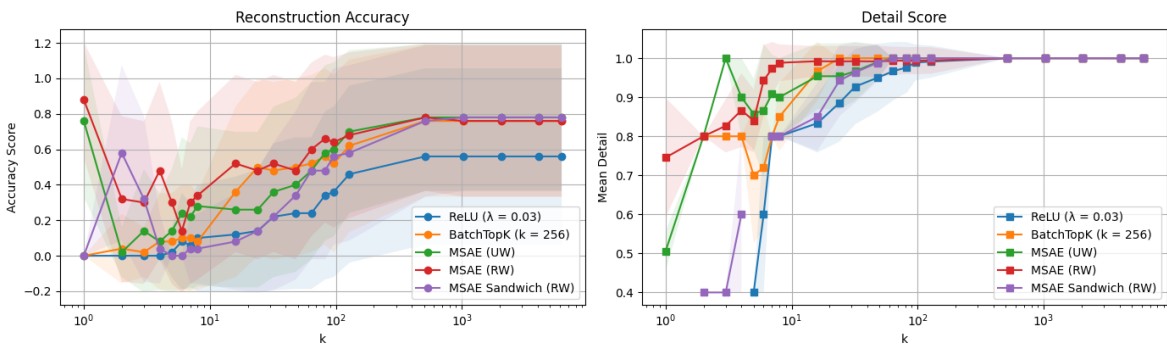

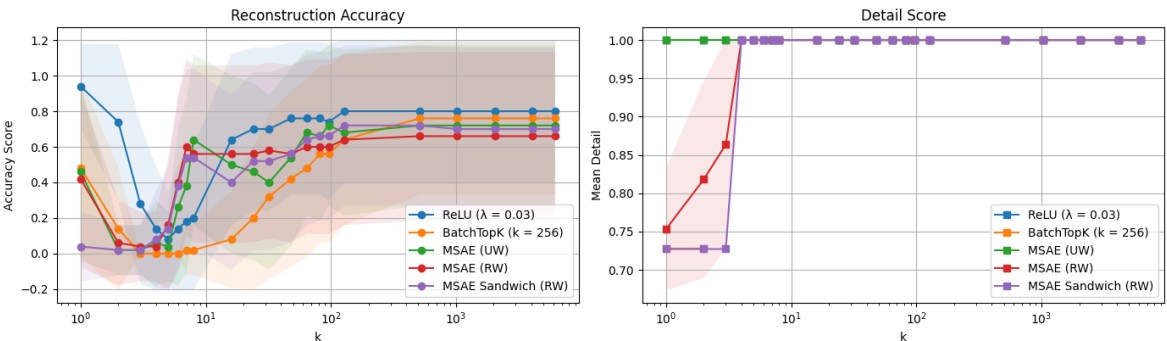

Below are specific image examples of the above classes.

**Hypernym path: entity -> physical_entity -> object -> whole -> living_thing -> organism -> animal -> chordate -> vertebrate -> reptile -> diapsid -> saurian -> lizard -> lacertid_lizard -> green_lizard**

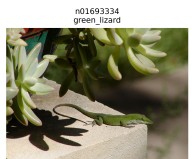

n01693334
green_lizard

|  | ReLU (λ = 0.03) | BatchTopK (k = 256) | MSAE (UW) | MSAE (RW) | MSAE Sandwich (RW) |
|---|---|---|---|---|---|
| k=1 | tailed_frog (off-path) | chordate (0.53) | chordate (0.53) | chordate (0.53) | chordate (0.53) |
| k=2 | tailed_frog (off-path) | whiptail (off-path) | teiid_lizard (off-path) | whiptail (off-path) | whole (0.27) |
| k=4 | teiid_lizard (off-path) | green_lizard (1.00) | teiid_lizard (off-path) | whiptail (off-path) | chordate (0.53) |
| k=8 | green_lizard (1.00) | green_lizard (1.00) | green_lizard (1.00) | green_lizard (1.00) | green_lizard (1.00) |
| k=16 | green_lizard (1.00) | green_lizard (1.00) | green_lizard (1.00) | green_lizard (1.00) | green_lizard (1.00) |
| k=32 | green_lizard (1.00) | green_lizard (1.00) | green_lizard (1.00) | green_lizard (1.00) | green_lizard (1.00) |
| k=64 | green_lizard (1.00) | green_lizard (1.00) | green_lizard (1.00) | green_lizard (1.00) | green_lizard (1.00) |
| k=128 | green_lizard (1.00) | green_lizard (1.00) | green_lizard (1.00) | green_lizard (1.00) | green_lizard (1.00) |
| k=512 | green_lizard (1.00) | green_lizard (1.00) | green_lizard (1.00) | green_lizard (1.00) | green_lizard (1.00) |
| k=2048 | green_lizard (1.00) | green_lizard (1.00) | green_lizard (1.00) | green_lizard (1.00) | green_lizard (1.00) |
| k=6188 | green_lizard (1.00) | green_lizard (1.00) | green_lizard (1.00) | green_lizard (1.00) | green_lizard (1.00) |

**Hypernym path: entity -> physical_entity -> object -> whole -> living_thing -> organism -> animal -> invertebrate -> worm -> flatworm**

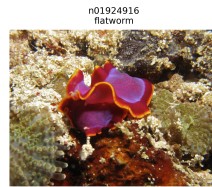

n01924916
flatworm

|  | ReLU (λ = 0.03) | BatchTopK (k = 256) | MSAE (UW) | MSAE (RW) | MSAE Sandwich (RW) |
|---|---|---|---|---|---|
| k=1 | eared_seal (off-path) | chordate (off-path) | invertebrate (0.80) | invertebrate (0.80) | chordate (off-path) |
| k=2 | chordate (off-path) | chordate (off-path) | hen (off-path) | sea_anemone (off-path) | whole (0.40) |
| k=4 | chordate (off-path) | american_coot (off-path) | sea_slug (off-path) | invertebrate (0.80) | chordate (off-path) |
| k=8 | chordate (off-path) | chordate (off-path) | sea_slug (off-path) | sea_slug (off-path) | ambystomid (off-path) |
| k=16 | sea_anemone (off-path) | sea_anemone (off-path) | sea_slug (off-path) | sea_anemone (off-path) | sea_anemone (off-path) |
| k=32 | flatworm (1.00) | sea_slug (off-path) | flatworm (1.00) | flatworm (1.00) | sea_anemone (off-path) |
| k=64 | flatworm (1.00) | flatworm (1.00) | flatworm (1.00) | flatworm (1.00) | flatworm (1.00) |
| k=128 | flatworm (1.00) | flatworm (1.00) | flatworm (1.00) | flatworm (1.00) | flatworm (1.00) |
| k=512 | flatworm (1.00) | flatworm (1.00) | flatworm (1.00) | flatworm (1.00) | flatworm (1.00) |
| k=2048 | flatworm (1.00) | flatworm (1.00) | flatworm (1.00) | flatworm (1.00) | flatworm (1.00) |
| k=6188 | flatworm (1.00) | flatworm (1.00) | flatworm (1.00) | flatworm (1.00) | flatworm (1.00) |

**Hypernym path: entity -> physical_entity -> object -> whole -> living_thing -> organism -> animal -> chordate -> vertebrate -> bird -> hen**

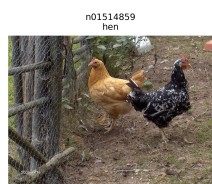

n01514859
hen

|  | ReLU (λ = 0.03) | BatchTopK (k = 256) | MSAE (UW) | MSAE (RW) | MSAE Sandwich (RW) |
|---|---|---|---|---|---|
| k=1 | hen (1.00) | oscine (off-path) | hen (1.00) | leatherback_turtle (off-path) | gallinaceous_bird (off-path) |
| k=2 | hen (1.00) | gallinaceous_bird (off-path) | gallinaceous_bird (off-path) | gallinaceous_bird (off-path) | gallinaceous_bird (off-path) |
| k=4 | gallinaceous_bird (off-path) | gallinaceous_bird (off-path) | gallinaceous_bird (off-path) | gallinaceous_bird (off-path) | gallinaceous_bird (off-path) |
| k=8 | gallinaceous_bird (off-path) | gallinaceous_bird (off-path) | hen (1.00) | hen (1.00) | hen (1.00) |
| k=16 | hen (1.00) | gallinaceous_bird (off-path) | hen (1.00) | gallinaceous_bird (off-path) | gallinaceous_bird (off-path) |
| k=32 | gallinaceous_bird (off-path) | gallinaceous_bird (off-path) | gallinaceous_bird (off-path) | gallinaceous_bird (off-path) | gallinaceous_bird (off-path) |
| k=64 | gallinaceous_bird (off-path) | gallinaceous_bird (off-path) | hen (1.00) | gallinaceous_bird (off-path) | gallinaceous_bird (off-path) |
| k=128 | hen (1.00) | gallinaceous_bird (off-path) | gallinaceous_bird (off-path) | gallinaceous_bird (off-path) | gallinaceous_bird (off-path) |
| k=512 | gallinaceous_bird (off-path) | hen (1.00) | hen (1.00) | gallinaceous_bird (off-path) | hen (1.00) |
| k=2048 | gallinaceous_bird (off-path) | hen (1.00) | hen (1.00) | gallinaceous_bird (off-path) | hen (1.00) |
| k=6188 | gallinaceous_bird (off-path) | hen (1.00) | hen (1.00) | gallinaceous_bird (off-path) | hen (1.00) |

### A.5 Sandwich Ablation

We compared four sampling variants: sandwich sampling with 1 and 2 random middle values, and random sampling with 3 and 4 random values. In order to assess which offers the best balance between computation and performance, we computed all metrics and the emissions.

| Model | $\hat{L}_0\uparrow$ | FVU $\downarrow$ | CS $\uparrow$ | LP (KL) $\downarrow$ | LP (Acc) $\uparrow$ | CKNNA $\uparrow$ | DO $\downarrow$ | NDN $\downarrow$ | HAG (%) $\downarrow$ |
|---|---|---|---|---|---|---|---|---|---|
| Sandwich (1) | $.785_{\pm.009}$ | $.004_{\pm.002}$ | $.998_{\pm.001}$ | $.003_{\pm.006}$ | $.988_{\pm.110}$ | $.722_{\pm.003}$ | **0.0021** | 1 | **12.6** |
| Sandwich (2) | $.789_{\pm.008}$ | $.004_{\pm.003}$ | $.998_{\pm.001}$ | $.003_{\pm.006}$ | $.987_{\pm.112}$ | $.721_{\pm.002}$ | **0.0021** | **0** | 14.3 |
| Random (3) | $.789_{\pm.009}$ | $.004_{\pm.003}$ | $.998_{\pm.001}$ | $.003_{\pm.007}$ | $.988_{\pm.109}$ | $.714_{\pm.004}$ | 0.0022 | **0** | 16.3 |
| Random (4) | $.793_{\pm.008}$ | $.004_{\pm.003}$ | $.998_{\pm.001}$ | $.003_{\pm.007}$ | $.987_{\pm.115}$ | $.714_{\pm.001}$ | 0.0022 | **0** | 16.7 |

Table 5: Quantitative comparison of MSAE models with sandwich sampling (1 and 2 middle values) and random sampling (3 and 4 random values). All models have almost equal performance, however Sandwich (1) has the least HAG score.

| Model | Duration (HH:MM:SS) | Emissions [$CO_2$eq] (kg) | Energy (kWh) | | | |
|---|---|---|---|---|---|---|
| | | | CPU | GPU | RAM | Total |
| Sandwich (1) | 01:32:49 | .082 | .044 | .247 | .015 | .307 |
| Sandwich (2) | 01:35:44 | .088 | .046 | .266 | .016 | .328 |
| Random (3) | 01:30:50 | .080 | .043 | .240 | .015 | .299 |
| Random (4) | 01:34:58 | .083 | .045 | .248 | .016 | .310 |

Table 6: Emissions during training MSAE sandwich and random models.

### A.6 Performance vs. Emissions

Table 7 provides a comprehensive comparison of the evaluated SAE variants across reconstruction quality, downstream performance, and environmental impact. MSAE (RW) and MSAE Sandwich (RW) secures the lowest fraction of variance unexplained (FVU = $.004_{\pm.003}$) and BatchTopK the highest neighborhood accuracy (CKNNA = $0.732_{\pm.003}$). Notably, the MSAE (RW) sandwich variant offers better performance than MSAE (RW) for lower cost. It cuts training carbon emissions by approximately 24% down to $0.082\,\mathrm{kg}$ of $CO_2$eq, while still maintaining highly competitive performance.

| Model | FVU ↓ | CKNNA ↑ | Emissions [$CO_2$eq] (kg) ↓ |
|---|---|---|---|
| ReLU ($\lambda = 0.03$) | $.293_{\pm.067}$ | $.626_{\pm.003}$ | **.064** |
| BatchTopK ($k = 256$) | $.023_{\pm.027}$ | $\mathbf{.732_{\pm.003}}$ | .067 |
| MSAE (UW) | $.077_{\pm.032}$ | $.727_{\pm.004}$ | .110 |
| MSAE (RW) | $\mathbf{.004_{\pm.003}}$ | $.705_{\pm.005}$ | .108 |
| MSAE Sandwich (RW) | $\mathbf{.004_{\pm.002}}$ | $.722_{\pm.003}$ | .082 |

Table 7: Performance vs. training emissions for all evaluated SAE model variants.

