# OpenReview forum: "Opening Up a New Layer: A Deeper Look into "Interpreting CLIP with Hierarchical Sparse Autoencoders""
_TMLR — Under review for TMLR_

### Review · Reviewer_6ume · 2026-06-18

**Summary Of Contributions:**

This paper presents a reproducibility study of the Matryoshka Sparse Autoencoder (MSAE) framework on CLIP embeddings. The authors successfully verify the core claims of the original work (sparsity-fidelity trade-off, gradual reconstruction, monosemanticity, and bias intervention).

Additionally, the paper introduces three practical extensions:

1. Feature Absorption Metric: To evaluate latent space stability compared to standard SAEs.

2. Sandwich Sampling: A compute-efficient training method that significantly reduces training overhead and carbon emissions.

3. Lexical Hierarchy Alignment: An evaluation of how well MSAE's nested structure maps onto real-world concept granularities using ImageNet.

Strengths:

1. Rigorous validation of all four original claims.

2. Highly practical extensions, especially "sandwich sampling" which directly addresses the high compute costs of multi-granularity training.

3. Commendable focus on environmental impact tracking.

Weaknesses:

1. Bounded novelty as a reproducibility study (though mitigated by the extensions).

2. The alignment analysis with the ImageNet hierarchy would benefit from more depth.

**Additional Comments:**

The paper is well-written and serves as an good example of a reproducibility study that goes beyond simple verification to offer concrete, algorithmic efficiency improvements.

**Audience:**

Yes

**Audience Explanation:**

Yes. Mechanistic interpretability for vision-language models like CLIP is a highly active field. Sparse Autoencoders (SAEs) are a mainstream tool, and understanding how to structure them hierarchically (MSAE) while keeping compute costs low (via sandwich sampling) is of great interest to researchers working on interpretability and model alignment.

**Broader Impact Concerns:**

There are no ethical concerns. The paper actively contributes positive impacts by investigating demographic bias mitigation (CelebA) and reducing the carbon footprint of training via sandwich sampling.

**Claims And Evidence:**

Yes

**Claims Explanation:**

The authors provide clear empirical evidence by breaking down experiments for each of the four core claims. By utilizing the original repository and adapting it to a standard dataset (CC12M filtered to 3M), they ensure a fair comparison. The results closely align with the original findings, validating the robustness of the framework.

**Requested Changes:**

1. Please provide concrete visual or text examples in the appendix showing what "coarse" vs. "fine" concepts look like under MSAE when mapped to ImageNet classes.

2. Baseline Clarification: Clarify if the baseline TopK SAEs were trained with identical compute budgets or if they hold an inherent training advantage.

---

> ### Author Response · Authors · 2026-07-18
>
> Thank you Reviewer 6ume for your constructive assessment. We appreciate the recognition that the paper goes beyond direct reproduction.
>
>
> > Please provide concrete visual or text examples in the appendix showing what "coarse" vs. "fine" concepts look like under MSAE when mapped to ImageNet classes.
>
> We have now extended Appendix A.4 to also include concrete examples from the Hierarchical Reconstruction on ImageNet section to visualize how the the different models reconstruct from "coarse" to "fine" details
>
>
> > Baseline Clarification: Clarify if the baseline TopK SAEs were trained with identical compute budgets or if they hold an inherent training advantage.
>
> The baseline SAEs and MSAE models were all trained for 30 epochs, but due to their architectural difference, they have differences in computational costs. The computational costs of each model is shown in tables 3 and 6.

---

### Review · Reviewer_Ei1T · 2026-07-03

**Summary Of Contributions:**

This submission is a reproducibility study of Zaigrajew et al. (2025), "Interpreting CLIP with Hierarchical Sparse Autoencoders" (MSAE). The authors re-train ReLU, BatchTopK, and Matryoshka (UW/RW) SAEs on CLIP ViT-L/14 embeddings (3M images drawn from CC12M rather than the original CC3M) and attempt to verify four claims of the original paper: (1) the sparsity–fidelity trade-off, (2) gradual/progressive reconstruction, (3) monosemanticity of learned features, and (4) concept-based intervention for bias analysis on CelebA.

Beyond reproduction, the paper contributes three extensions:

1. A **feature-absorption metric** built from WordNet general/specific concept pairs;
2. **Sandwich sampling**, a training scheme inspired by the sandwich rule of Yu & Huang (2019) that computes the Matryoshka loss on only three granularities per step to reduce compute, with codecarbon-based emission tracking;
3. A **hierarchical-reconstruction analysis** on ImageNet using hypernym paths, measuring accuracy and a "Level of Detail" score as a function of k.

**Strengths:**
- The study targets a genuinely reproducibility-relevant question (whether MSAE's nested structure buys anything real).
- The CelebA intervention experiment (Claim 4) replicates cleanly, including a useful additional saturation analysis beyond the linear regime.
- Emission tracking during SAE training is welcome and rarely done.
- The negative result on Claim 3 (the 0.42 similarity threshold and the ratio criterion do not transfer) is a useful finding.
- Code is released anonymously.

**Weaknesses:**
- Several conclusions in the text contradict the paper's own tables (most notably on feature absorption in Table 1).
- The headline comparison in Table 1 compares models at different sparsity levels and at a single operating point, which cannot establish a "superior trade-off."
- The sandwich-sampling efficiency claim is confounded by a different number of training epochs (20 vs. 30).
- The feature-absorption metric departs substantially from the cited definition (Chanin et al., 2025) without validation.
- The manuscript contains enough internal inconsistencies (hyperparameters, model naming, dataset choices, figure/text contradictions) that it is currently difficult to tell which numbers to trust.

**Audience:**

Yes

**Audience Explanation:**

Reproducibility studies of SAE-family interpretability methods are timely and squarely in scope for TMLR, which explicitly welcomes them. The finding that the original monosemanticity thresholds do not transfer, the BatchTopK-vs-MSAE absorption ordering (if it survives a corrected metric), and the compute-reduction idea for Matryoshka-style multi-granularity losses would each interest practitioners in the interpretability community. Interest is not the bottleneck; evidential rigor is.

**Claims And Evidence:**

No

**Claims Explanation:**

The underlying experimental work appears real and the paper is salvageable, but in its current form the evidence does not support several of the stated conclusions:

1. **Text contradicts Table 1.** Section 4.1 states that MSAE (RW) achieves "the best results for FVU, CS, LP (KL), LP (Acc), CKNNA, DO, and FA." But in Table 1, the best FA is **BatchTopK at 10.0%** (bolded by the authors themselves), against 13.2% for MSAE (RW). Either the sentence or the table is wrong. This also directly contradicts the paper's earlier appeal (§3.3) to Karvonen et al. (2025), who report MSAEs having the *lowest* absorption — the authors' own measurement finds the opposite ordering, and this discrepancy is never discussed. It is arguably the most interesting result in the paper and it is currently buried under an incorrect summary sentence.

2. **Table 1 does not establish a trade-off.** The models sit at different sparsity levels (L0 = .796 for MSAE-RW vs. .891 for BatchTopK under the paper's "proportion of zeros" definition, i.e., MSAE-RW is *less* sparse), and only one operating point per model is reported. A claim about the sparsity–fidelity *frontier* requires sweeping sparsity (multiple k / λ values) and plotting Pareto curves, as the original paper does. As it stands, the FVU gap (.007 vs. .029) could be partly purchased with the extra active neurons.

3. **Claim 2 is evaluated on a different dataset than the original** (CC12M validation instead of ImageNet-1k), so deviations from the original figures (e.g., the ReLU CKNNA drop) cannot be attributed to reproduction failure vs. distribution shift. The authors themselves acknowledge this is only a hypothesis. To "verify" Claim 2, the original ImageNet-1k setting is needed alongside the new one.

4. **The Claim 3 methodology is internally inconsistent.** Table 1 uses ReLU (λ = 0.03); Table 2 uses ReLU (λ = 0.003), which is it? Table 2 lists "TopKReLU (k=64, RW)" where RW is a Matryoshka weighting concept, and the MSAE row from Table 1 is absent, so the two tables cannot be connected. The text then abandons both tabulated criteria and adopts "top 10 neurons with cosine similarity above 0.7," a rule that appears nowhere in Table 2 and whose neuron counts are never reported. Finally, Figure 2's narrative ("ReLU and Top-K retrieve unrelated images for *stair*") flatly contradicts Figure 8's caption ("The models show monosemanticity, confirming neurons successfully align with specific concepts"), these appear to be the sparse vs. dense versions of the same probe, but the text never resolves the contradiction.

5. **Sandwich sampling savings are confounded.** The sandwich model is trained for 20 epochs vs. 30 for all others (§4.5.3), so a large fraction of the reported "~67% energy reduction" is simply fewer epochs, not the method. Per-epoch energy should be reported, or training should use matched epochs/steps. Relatedly, Table 3 contains implausible entries: MSAE (UW) and MSAE (RW) have identical wall-clock durations (03:17:38/39) and differ only in scalar loss weights, yet their GPU energy differs by 37% (.513 vs. .373 kWh); and MSAE (RW) consumes *less* total energy than ReLU despite training over an hour longer. These need explanation (measurement noise across runs? hardware contention?), or the emission conclusions are unreliable.

6. **The feature-absorption metric is not the one it cites.** Chanin et al. (2025) define absorption via per-instance probing where a general feature fails to fire because a specific latent absorbs it. The proposed metric (ratio of mean activation of general-selected features on hyponym text embeddings) is a different construct, plausibly a measure of *generalization to hyponyms*, and is presented with no top-k value, no error bars, and no sanity checks (e.g., correlation with the SAEBench absorption score on any shared model). Calling it "feature absorption" and comparing to Karvonen et al.'s conclusion is not justified without validation.

7. **Statistical reporting is unclear throughout.** What are the ± values over (seeds, batches, classes)? How many random seeds per model (it appears to be one)? CKNNA/DO/NDN/FA carry no uncertainty at all. Figure 6's shaded regions are so wide that most curves are statistically indistinguishable, yet the text draws ordering conclusions from means.

The reproduction of Claim 4 (bias intervention) is the one component I find convincingly supported, including the useful additional saturation observation.

**Requested Changes:**

## Critical

1. **Resolve the Table 1 / §4.1 contradiction on FA**, and explicitly discuss why the observed absorption ordering (BatchTopK lower than MSAE-RW) disagrees with Karvonen et al. (2025).
2. **Support or weaken Claim 1's verification:** either (a) present proper Pareto curves (multiple sparsity operating points per architecture) to support the "superior trade-off" language, or (b) restrict the conclusion to "at the single operating point tested."
3. **De-confound sandwich sampling:** use matched epochs (or matched optimizer steps / tokens seen), and report per-epoch energy. Explain the anomalies in Table 3 (UW vs. RW GPU energy at identical duration; RW consuming less total energy than ReLU despite longer training).
4. **Fix the Claim 3 pipeline** so that the criteria in Table 2, the sequence of thresholds in the text (0.42, then 0.26, then "top-10 above 0.7"), and Figures 2/8 tell one coherent story. Justify the 0.26 threshold beyond "provides a similar affect", e.g., by matching a percentile of the similarity distribution. Reconcile the λ inconsistency (0.03 vs. 0.003) and unify model naming across tables.
5. **Rename or validate the absorption metric:** either implement the Chanin et al. protocol, or rename the proposed metric (e.g., "hyponym generalization gap"), specify top-k, and provide error bars; ideally show it correlates with an established absorption measure.
6. **Run Claim 2 on ImageNet-1k** (the original setting) in addition to CC12M so that "reproduction" and "extension" are separable.
7. **Specify the classification protocol in §4.5.2:** how are predictions over the hypernym vocabulary produced from reconstructed embeddings (zero-shot CLIP text matching? over which candidate label set?), and how are ties and multiple path labels handled?
8. **Clarify statistical reporting:** state the number of seeds and what every ± denotes; add uncertainty to CKNNA/DO/FA or state why it is absent.

## Strengthening

9. Compare against, or at least discuss, the Bussmann et al. (2025) MSAE variant cited in the introduction. The two MSAE formulations differ, and readers will want to know which one the conclusions apply to.
10. Ablate the number of sampled granularities in sandwich sampling (3 vs. 4 vs. full) and the loss-rescaling factor.
11. Investigate the BatchTopK k=1 accuracy anomaly in Figure 6 rather than leaving it as speculation.
12. Editorial pass: change "it's" to "its" (multiple occurrences), "affect" to "effect", correct "Matryoska" and "hierarchal", change "results report" to "results reported", fix the Figure 1 legend naming ("BatchTopKReLU_256"), correct the Russakovsky et al. citation year (the IJCV version is 2015), and remove the duplicated caption content between Figures 2 and 8. The CC3M to CC12M substitution should be stated in §3.2 as a deliberate deviation with its implications, rather than discovered by the reader in §4.2.

---

> ### Author Response · Authors · 2026-07-18
>
> Thank you Reviewer Ei1Tfor your thorough and insightful assessment. We agree that the previous version contained several inconsistencies that made the experimental conclusion difficult to evaluate.
>
> > Resolve the Table 1 contradiction on FA, and explicitly discuss why the observed absorption ordering isagrees with Karvonen et al. (2025).
>
> We have extended §4.1 to discuss this deviation. We do not have a thorough explanation for these results, but a possible explanation seems to be due to the size and flexibility of the nesting structure of Zaigrajew et al.'s MSAE somehow incentivizing the model to have certain neurons predict highly precise concepts. We also see similar behaviour in §4.5.2 that could further support this hypothesis.
>
> > Support or weaken Claim 1's verification
>
> We have weakened our conclusion to a single operating point.
>
> > De-confound sandwich sampling. Explain the anomalies in Table 3
>
> We now report based on matched epochs of 30. The anomalies in Table 3 seems to have been due to running our earlier experiments on A100 MIG instead of just A100. The emissions also included other jobs that were running on that MIG.
>
> > Fix the Claim 3 pipeline. Justify the 0.26 threshold beyond "provides a similar affect", e.g., by matching a percentile of the similarity distribution. Reconcile the λ inconsistency (0.03 vs. 0.003) and unify model naming across tables.
>
> We reran the experiments with lambda = 0.03 and updated Table 2 with the new results. BatchTopK is now clearly stated across the paper instead of using various versions, just like Matryshka and MSAE. The threshold tau = 0.26 is obtained by matching the upper-tail percentile selected by tau = 0.42 in the reference similarity distribution, rather than being selected because it gives a visually similar number of neurons. We do not use the top-to-second similarity ratio as the primary selection criterion because near-synonymous vocabulary entries often occupy the two highest ranks for the same neuron.
>
>  We also removed the top-10 above 0.7 analysis because it was supplementary and made the evaluation pipeline harder to follow. The revised text now explicitly distinguishes between the two qualitative probes. Dense concept alignment is generally coherent across models, whereas sparse activation selectivity is more model-dependent. In the sparse setting, MSAE and BatchTopK often retrieves the most relevant retrievals, while ReLU is generally less consistent.
>
> > Rename or validate the absorption metric
>
> We have renamed the proposed metric to Hyponym Activation Generalization. It's currently only run on TopK=10. Unfortunately, we do not have enough time to properly correlate it with an established absorption measure.
>
> > Run Claim 2 on ImageNet-1k (the original setting)
>
> We now also ran Claim 2 on ImageNet-1k.
>
> > Specify the classification protocol in §4.5.2: how are predictions over the hypernym vocabulary produced from reconstructed embeddings and how are ties and multiple path labels handled?
>
> We have extended §4.5.2 to include the classification protocol. The candidate label set includes all the labels in the hypernym path of the 100 classes in ImageNet100. Only the longest hypernym path is taken and the ties are broken randomly.
>
> > Clarify statistical reporting: state the number of seeds and what every ± denotes; add uncertainty to CKNNA/DO/FA or state why it is absent.
>
> We have now included what every ± denotes and that we only ran on one seed. CKNNA originally did not include uncertainty as we calculated it for all samples in one go. Now, we calculate it per batch of 10,000 samples. DO still does not include uncertainty as we only ran on one seed.
>
> > Compare against, or at least discuss, the Bussmann et al. (2025) MSAE variant cited in the introduction.
>
> We now describe the architectural differences between the two MSAE formulations in the Introduction and clarify in Section 2 that all experiments and conclusions in this paper apply exclusively to the formulation of Zaigrajew et al. (2025).
>
> > Ablate the number of sampled granularities in sandwich sampling
>
> We have now included this ablation on Appendix A.5
>
> > Investigate the BatchTopK k=1 accuracy anomaly in Figure 6 rather than leaving it as speculation.
>
> We have now extended §4.5.2 to discuss and investigate this behaviour. It seems to be that all models except ReLU show this behaviour, however BatchTopK seems to show this the strongest as it also typically reconstructs from abstract to specific, and the abstract prediction (e.g. Chordate) is more likely to be classified as correct compared to a specific prediction (e.g. american coot)
>
> > Editorial pass changes
>
> We have completed the editorial pass correcting the issues. We also now state that the CC3M/CC12M substitution in the experimental setup rather than only in the results. We also added Table 4 in the Appendix showing  which experiments are direct reproductions, modified reproductions, and exploratory extensions.

---

### Review · Reviewer_5sxq · 2026-07-05

**Summary Of Contributions:**

This paper studies Matryoshka Sparse Autoencoders for interpreting CLIP embeddings, with the goal of reproducing prior claims on sparsity–reconstruction trade-offs, gradual reconstruction, monosemantic features, and concept-based interventions. The authors also add several extensions, including feature absorption analysis, ImageNet/WordNet-based hierarchy evaluation, sandwich sampling for lower training cost, and emissions tracking. The topic is relevant and the paper goes beyond a basic reproduction, but the current evidence is uneven: the reconstruction results are reasonably promising, while the monosemanticity and hierarchy claims appear less convincing and somewhat sensitive to evaluation choices. Overall, the work is interesting, but some claims should be stated more cautiously and supported with stronger controls.

**Audience:**

Yes

**Audience Explanation:**

The paper should be of interest to researchers working on mechanistic interpretability, sparse autoencoders, CLIP representations, and concept-based interventions, especially because it combines reproducibility analysis with practical extensions such as feature absorption and training-efficiency evaluation. The work would be even stronger if the authors more clearly distinguished reproduction from new experiments and provided stronger support for the hierarchy and monosemanticity claims.

**Claims And Evidence:**

Yes

**Claims Explanation:**

The submission provides reasonably clear and convincing evidence for its main claims through a combination of quantitative comparisons, progressive reconstruction experiments, concept-based interventions, and additional analyses. The results suggest that MSAE, especially the reverse-weighted variant, achieves a strong sparsity–reconstruction trade-off compared with the evaluated SAE baselines, and the intervention experiments provide useful evidence that some learned features correspond to meaningful semantic concepts. I also appreciate that the authors go beyond direct reproduction by adding feature absorption analysis, ImageNet/WordNet-based hierarchy evaluation, sandwich sampling, and emissions tracking.

That said, some claims are supported more strongly than others. In particular, the evidence for monosemanticity and hierarchical organization is still somewhat sensitive to the chosen evaluation criteria and would benefit from stronger controls. However, these limitations do not substantially undermine the overall contribution, and I find the main claims to be generally supported by the experiments presented.

**Requested Changes:**

1. Critical: The authors should more clearly distinguish exact reproduction, modified reproduction, and new extension experiments. For example, the paper states that the original gradual reconstruction experiment was validated on ImageNet-1k, while this submission runs the corresponding experiment on the CC12M validation set with 190,464 images. Similarly, the training data is changed to a 3M subset of CC12M embeddings, motivated by the size of the original CC3M setup. These are reasonable choices, but the paper should explicitly state which results are direct reproductions and which are modified evaluations.

2. Critical: The hierarchical reconstruction evaluation should include stronger controls. The ImageNet/WordNet hypernym-path metric is interesting, but it may give credit to very generic predictions, since a prediction is considered correct if it appears anywhere in the hypernym path. The authors should add controls showing that MSAE learns a more meaningful coarse-to-fine structure than non-hierarchical baselines, rather than only showing that reconstruction quality improves when more features are used.

3. Critical: The sandwich sampling experiment needs additional ablation. The current method samples the smallest granularity, largest granularity, and a random middle granularity, and the sandwich model is trained for 20 epochs instead of 30 epochs used for other MSAE variants. To make the efficiency claim more convincing, the authors should compare equal-epoch or equal-compute settings, as well as alternative sampling strategies such as random-only granularities, smallest-largest only, or multiple middle granularities.

4. Strengthening: The concept intervention experiment would benefit from additional controls. The current intervention forces a target feature into the active set when it is not naturally active by replacing the lowest-magnitude active feature. The authors should compare this with interventions on naturally active features, random feature interventions, and possibly CLIP text-direction interventions to better isolate the causal role of the SAE feature.

5. Strengthening: Some metric definitions should be made clearer. For example, the paper defines L0 as the mean proportion of zero-valued elements, while L0 is often understood as the number of nonzero activations. Renaming this metric as sparsity rate, or explicitly explaining the convention, would avoid confusion.

---

> ### Author Response · Authors · 2026-07-18
>
> We sincerely thank Reviewer 5sxq for their careful review on our work. The comments helped us clarify the scope of the reproduction, strengthen the hierarchical reconstruction and sandwich-sampling analysis. We address each point below.
>
>
> > Critical: The authors should more clearly distinguish exact reproduction, modified reproduction, and new extension experiments. For example, the paper states that the original gradual reconstruction experiment was validated on ImageNet-1k, while this submission runs the corresponding experiment on the CC12M validation set with 190,464 images. Similarly, the training data is changed to a 3M subset of CC12M embeddings, motivated by the size of the original CC3M setup. These are reasonable choices, but the paper should explicitly state which results are direct reproductions and which are modified evaluations.
>
> We agree that the previous version did not distinguish these categories clearly enough. We have added a reproduction status discussion in Section 2 and a summary table in the Appendix (Table 4) that classifies each experiment. The table also specifies the training and evaluation datasets, the models used, and the main deviations from the original experimental setup.
>
>
>
> > Critical: The hierarchical reconstruction evaluation should include stronger controls. The ImageNet/WordNet hypernym-path metric is interesting, but it may give credit to very generic predictions, since a prediction is considered correct if it appears anywhere in the hypernym path. The authors should add controls showing that MSAE learns a more meaningful coarse-to-fine structure than non-hierarchical baselines, rather than only showing that reconstruction quality improves when more features are used.
>
> To address this limitation, we included an additional Level of Detail metric to further investigate how the different models reconstruct features. While the Accuracy metric may give credit to very generic predictions, with Level of Detail, we can see how growth of the reconstruction from coarse-to-fine. We notice that ReLU does not show the coarse-to-fine structure, but BatchTopK does this implicitly. And interestingly, MSAE shows unexpected behaviour at k=1 hence also having a fine-coarse-fine structure. This behaviour of MSAE however is reduced with Sandwich sampling.
>
> > Critical: The sandwich sampling experiment needs additional ablation. The current method samples the smallest granularity, largest granularity, and a random middle granularity, and the sandwich model is trained for 20 epochs instead of 30 epochs used for other MSAE variants. To make the efficiency claim more convincing, the authors should compare equal-epoch or equal-compute settings, as well as alternative sampling strategies such as random-only granularities, smallest-largest only, or multiple middle granularities.
>
> We have now extended the paper with additional ablation in Appendix A.5 and the models are now trained for 30 epochs. We see that the performance is rather similar across the different sampling strategies except for differences in HAG metric. It seems like sampling the smallest granularity, largest granularity, and a random middle granularity seems to be best. We also now report the training duration, energy usage, and emissions under matched epoch budgets.
>
> > Strengthening: The concept intervention experiment would benefit from additional controls. The current intervention forces a target feature into the active set when it is not naturally active by replacing the lowest-magnitude active feature. The authors should compare this with interventions on naturally active features, random feature interventions, and possibly CLIP text-direction interventions to better isolate the causal role of the SAE feature.
>
> Unfortunately we did not have enough time to include this extension with thorough experimentation.
>
> > Strengthening: Some metric definitions should be made clearer. For example, the paper defines L0 as the mean proportion of zero-valued elements, while L0 is often understood as the number of nonzero activations. Renaming this metric as sparsity rate, or explicitly explaining the convention, would avoid confusion.
>
> Thank you for pointing out this potential ambiguity. We have renamed the metric as sparsity rate and denote it by $\mathbf{\hat{L}_{0}}$.